# Sliding Cutting and Cutting Parameters of Concentric Curvilineal Edge Sliding Cutter for *Caragana korshinskii* (C.K.) Branches

Haifeng Luo [1,2,*], Shaojun Guo [1,2], Zhenkun Zhi [1,2] and Jiangming Kan [1,2]

1   School of Technology, Beijing Forestry University, Beijing 100083, China; guoshaojun@bjfu.edu.cn (S.G.); zzk7210279@bjfu.edu.cn (Z.Z.); kanjm@bjfu.edu.cn (J.K.)
2   Key Lab of State Forestry Administration on Forestry Equipment and Automation, Beijing Forestry University, Beijing 100083, China
*   Correspondence: luohaifeng@bjfu.edu.cn; Tel.: +86-010-62338144

**Abstract:** To realize the reduction in cutting force and guarantee pruning section quality in the pruning and stubble work of *Caragana korshinskii* (C.K.), a concentric curvilineal edge sliding cutter was proposed and the related cutting characteristics were studied. The impacts of branch diameter (D), cutting speed ($V_c$), blade wedge angle ($\beta$), cutting clearance (c) and moisture content (W) on peak torque (T) and cutting energy (E) with this cutter were explored in single-factor tests. On the basis of the Box—Behnken principle, a multi-factor test was further conducted based on the single-factor tests with $V_c$, $\beta$ and c as influencing factors and with T and E as targets, and a regression model was established. Test results indicate that the peak torque (T) increases with the increase in D and $\beta$ and reduces with the growth of $V_c$ and W; with the increase in c, it reduces first and then rises; the cutting energy (E) increases with the growth of D and $\beta$, declines with the increase in W and diminishes first and then rises with the increase in $V_c$ and c. The optimal parameter combination of the regression model was obtained with $V_c$ of 2.16 rad/s, $\beta$ of 20° and c of 1.0 mm, which resulted in a T of 17.25 N·m and P of 7.03 J. The discrepancies between the observed and forecasted values for T and E are 0.87% and 5.004%. New cutting tool and data support for the development of subsequent C.K. branch stubble equipment can be obtained with this new sliding cutter.

**Keywords:** *Caragana korshinskii* branches; reduction in cutting force; section quality; concentric curvilineal edge; sliding cutter; optimal parameter combination

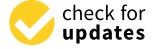



## 1. Introduction

*Caragana korshinskii*, commonly known as C.K., is a small shrub that inhabits sandy areas across northwest China. Its widespread distribution makes it a prominent plant species in the region. It possesses qualities such as drought and heat resistance, windbreak and sand stabilization abilities, soil and water conservation properties and strong vigor [1]. In accordance with the biological characteristics and planting agronomic requirements of C.K., it is necessary to execute stubble pruning every 3 to 5 years. Stubble pruning is an essential measure for the rejuvenation of C.K. shrub vegetation. If the stubble pruning operation is not carried out, the phenomenon of withering and declining will appear, and its natural mortality will also increase [2]. Pruning stubble at a flat angle is also very important, as it will directly affect the growth after stubble. In many areas, degradation is caused by uneven or residual stubble, which puts planted areas at risk of secondary desertification. Therefore, research into new cutting tools and harvesting equipment is vital to the development of the desert shrub industry [2].

The cutting method of shrubs mainly refers to the direction of the cutter entering the material, which is subdivided into two basic models: straight-across cutting and sliding cutting. Straight-across cutting refers to the cutting method in which the absolute motion direction of the cutter is perpendicular to the cutting edge of the cutter. Sliding cutting

refers to the cutting method in which the absolute motion direction of the cutter is neither vertical nor parallel to the cutting edge of the cutter. The mechanical experimental results of the cutting theory and the geometric analysis results of the cutter motion show that sliding cutting is a more efficient method than straight-across cutting. It reduces the labor required and improves overall output [3]. The cutting types of shrubs are divided into supported cutting and unsupported cutting. Supported cutting is to apply a supporting force in the opposite direction of the moving blade movement. Cutting using a moving blade with a fixed blade is called single-support cutting. Supported cutting can make the branch obtain a certain bending resistance, and can be cut at low speed, which is beneficial to the cutting of shrubs. Unsupported cutting only has a moving blade to cut the shrub, which is cut in the absence of any support for the shrub. It is necessary to obtain a large inertia force through high speed. The greater the speed, the stronger the bending resistance, which is conducive to the cutting of the shrub [3]. The choice of cutting method directly affects the cutting quality and working efficiency of the cutting equipment [4]. At present, the cutting methods used in forestry and agricultural harvesting equipment are mainly divided into the following four types: reciprocating, chain sawing, flailing and sawing [5]. After a great many practical tests and analysis, the characteristics of the four cutting methods are obtained: (1) A reciprocating cutting tool is a single-support cutter, which has good cutting effect, low power consumption and is widely used in wheat harvesters and lawn mowers. However, the tool needs to change direction continuously and has inertial impact [6]. (2) Chain saw cutting tools are fast, efficient, portable, and easy to carry. They are mainly applied to working conditions such as wood cutting and tree cutting. In the actual cutting process, chains often jam and the tools cannot work properly due to interference from leaves and branches [7]. (3) When the flailing tool is used to cut a branch with a smaller diameter, the cutting efficiency is high and the effect is good [8]. (4) The sawing tool is an unsupported cutting type, and the cutting section is neat and tidy, but the circular saw blade needs to strictly control the cutting parameters during the working process. Otherwise, there will be branch splitting, burrs and even large area burns, and the cutting energy is large [9–11].

Currently, numerous scholars have conducted extensive research on crop stem cutting mechanism, motion analysis and mechanical properties [12,13]. According to the characteristics of sisal, Song et al. proposed a method of rotating cutting of sisal, and tried many different cutting parameters to explore which cutting parameters have a significant correlation with the mechanical properties of sisal leaves. According to their experimental results, the cutting speed has a great influence on the ultimate shear stress and specific cutting energy of sisal leaves. The minimum cutting shear stress can be achieved when the cutting speed exceeds a specific value. In addition, when the blade slope is 40°, the cutting section quality is better. The proposed cutting method presented in this paper proves to be efficient and serves as valuable reference for the advancement of sisal harvesting equipment [14]. A reciprocating cutting test bench was used by Zhang et al. Six influencing factors, including cutting inclination angle, blade inclination angle, average cutting speed, feed speed, millet area and blade combination, were adopted to carry out single-factor experimental research. Results indicate that the average cutting speed, cutting inclination angle and blade inclination angle have a great influence on the cutting force. The optimal parameter combination was found and verified by experiments [15]. To explore the shear mechanical properties of apple branches and find the optimal combination of cutting parameters, Kang et al. carried out shear testing of apple branches on a reciprocating cutter test bench developed by themselves. The test methods were single-factor test and multi-factor test. The single-factor test took the branch diameter, average cutting speed, cutting gap and tool sliding angle as the influencing factors to explore the influence of these four factors on the peak cutting force of the branch. On this basis, the tool slip angle, cutting clearance and the mean of cutting speed served as the influencing factors, and the relationship between them and the peak cutting force was explored by multi-factor experiment, and a regression model was built. After analyzing the regression model, the

optimal cutting parameter combination was obtained. The optimized results were true and reliable, which provided reliable data and reference for the development of subsequent apple branch-related cutting equipment [16]. The cross-section quality not only affects the germination of C.K. in the second year, but demonstrates the substantial mechanical attributes of the stubble. In order to diminish the power consumed by cutting and improve the quality of the cross section, researchers carried out branch sawing tests on a test bench developed by themselves. The star point design test method was adopted, and the moving speed, sawing speed and blade inclination angle were selected as the influencing factors. Section integrity rate and sawing power were selected as the target values. After adjusting the working parameters, a mathematical regression model was built and the optimal parameter combination was acquired and verified [17]. To solve the problems of serious wear of saw blade and poor cutting effect of sawing surface caused by unreasonable working parameters in sawing, a self-developed branch sawing test bench was used for experimental research. The experimental method was a single-factor test. Six influencing factors of branch diameter, sawing angle, sawing speed, feed speed, moisture content and circular saw blade teeth were selected. On this basis, a multi-factor test was performed. The sawing speed, feed speed, sawing angle and the number of circular saw teeth were taken as the influencing factors, and the sawing power consumption and sawing surface quality were taken as the target values. Finally, a mathematical regression model was obtained. Through the analysis of the regression model, the best parameter combination was found. These research data provide data for the follow-up progression of C.K. stumping props and harvesting equipment [18].

Based on the mentioned research, a concentric curvilineal edge sliding cutter is developed; sliding cutting, the reduction in cutting force and the quality of the cutting section are considered. Cutting tests of branches were carried out with this new cutter on the self-developed cutting test bench. The effects of test factors on the target value were explored through single-factor and multi-factor tests to find the optimal combination of cutting parameters.

## 2. Materials and Methods

### 2.1. Test Materials

The material for this experiment was collected from Yanchi, Ningxia ($37°59'48''$ N, $107°2'38''$ E), in May 2023, at the age of 3 years (Figure 1). The selected branches were essentially straight, without nodes, pests and diseases, ranging from 4 mm to 14 mm in diameter and 150 mm in length. They had good toughness and abrasion resistance. The branches used in the experiment were packaged in sealed bags and refrigerated to prevent changes in the internal moisture content. The moisture content was measured within 24 h of the C.K. cutting test. Cr12MoV, with good toughness and wear resistance, was selected as tool material. According to GB/T1299-2014 [19], the chemical composition of Cr12MoV is C 1.45%~1.70%, Si $\leq$ 0.40%, Mn $\leq$ 0.4%, Cr 11.0%~12.5%, Mo 0.40%~0.60%, V 0.15%~0.30%, S $\leq$ 0.03%, $p \leq$ 0.03%. The heat treatment process of Cr12MoV was to heat the material in the range of 950~1000 °C, followed by oil-cooled quenching for 20 min. After that, it was tempered twice at 510~520 °C to achieve a material hardness value $\geq$ 60 HRC.

### 2.2. Cutting Test Bench

The overall structure of the self-developed cutting test bench is shown in Figure 2a, which mainly consists of four parts: the concentric curvilineal edge sliding cutter, the drive system, the control and measurement system and the feeding system.

The concentric curvilineal edge sliding cutter consists of two cutterheads, curvilineal edge blades, chains and chain wheels. The curvilineal edge blades are mounted on the cutterheads. The cutterheads are attached to the driven chain wheels. The drive system consists of two stepper motors, three couplings, shafts and drive chain wheels. The rotation and torque of the step motors are transformed to the cutterheads with two horizontal chain links. The control and measurement system consists of the motion control card, the

dynamic torque sensor and the laptop. The motors are controlled by the laptop through the motion control card and the real-time torque data from the torque sensor assembled on the drive chain wheel shaft are transferred to the laptop via the serial port. The feeding system is divided into five parts: the branch holder plate, the moving guide rail, the servo motor and two limit sensors. The branch holder plate is mounted on the moving guide rail; the moving guide rail is driven by the motor and is limited by two limit sensors.

### 2.3. Test Indicators

### 2.3.1. Peak Torque

During shrub harvesting operations, the peak cutting force is an important parameter affecting the cutting performance [20–22]. Since the concentric curvilineal edge sliding cutter is in the form of a rotating cutterhead, a torque sensor is added between the motor shaft and the actual working shaft to acquire real-time torque. By collecting and processing the real-time torque information, the peak torque value is selected as the target value to measure the cutting effect. With the actual cutting radius being measured, the peak torque value can be converted to peak cutting force to compare the cutting with other tools.

### 2.3.2. Cutting Energy

The expenditure of energy is a significant parameter, which is vital to promote the harvesting equipment and prolong the working time of harvesting equipment [5,23]. In this paper, real-time torque information is collected and stored and the cutting energy is acquired by handling the real-time torque data [24] (Equation (1)):

$$\mathrm{E} = \omega \cdot \int T(t)dt \tag{1}$$

where E is the energy, J; $\omega$ is the rotational speed, rad/s; $T(t)$ is variation curve of torque data with time; $t$ is the time variable, s.

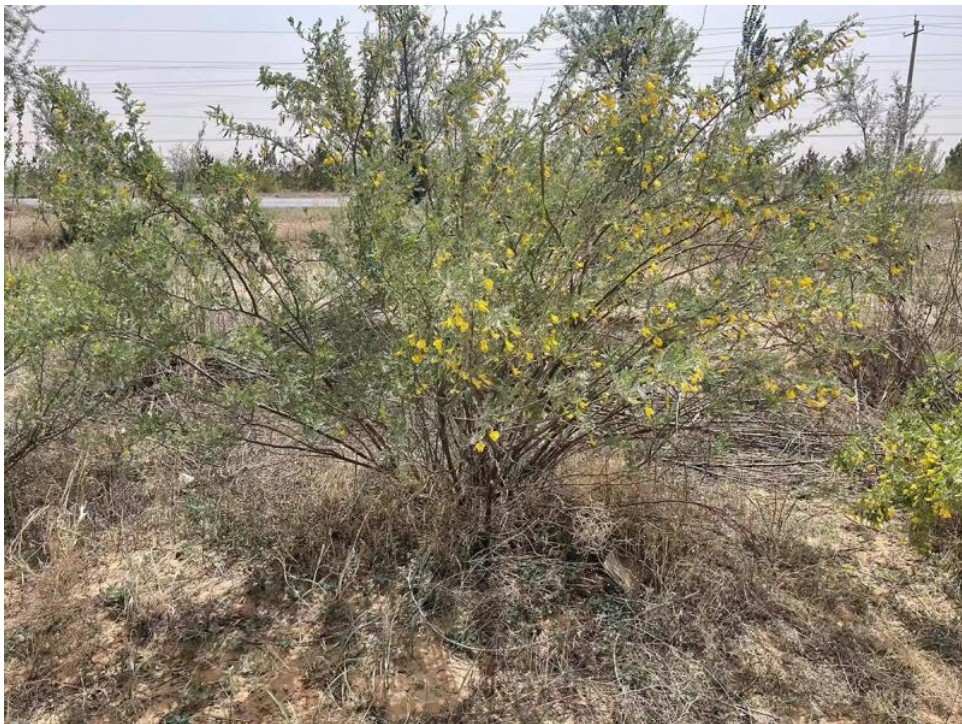

**Figure 1.** Three-year-old C.K. in Yanchi County, Wuzhong City, Ningxia Hui Autonomous Region.

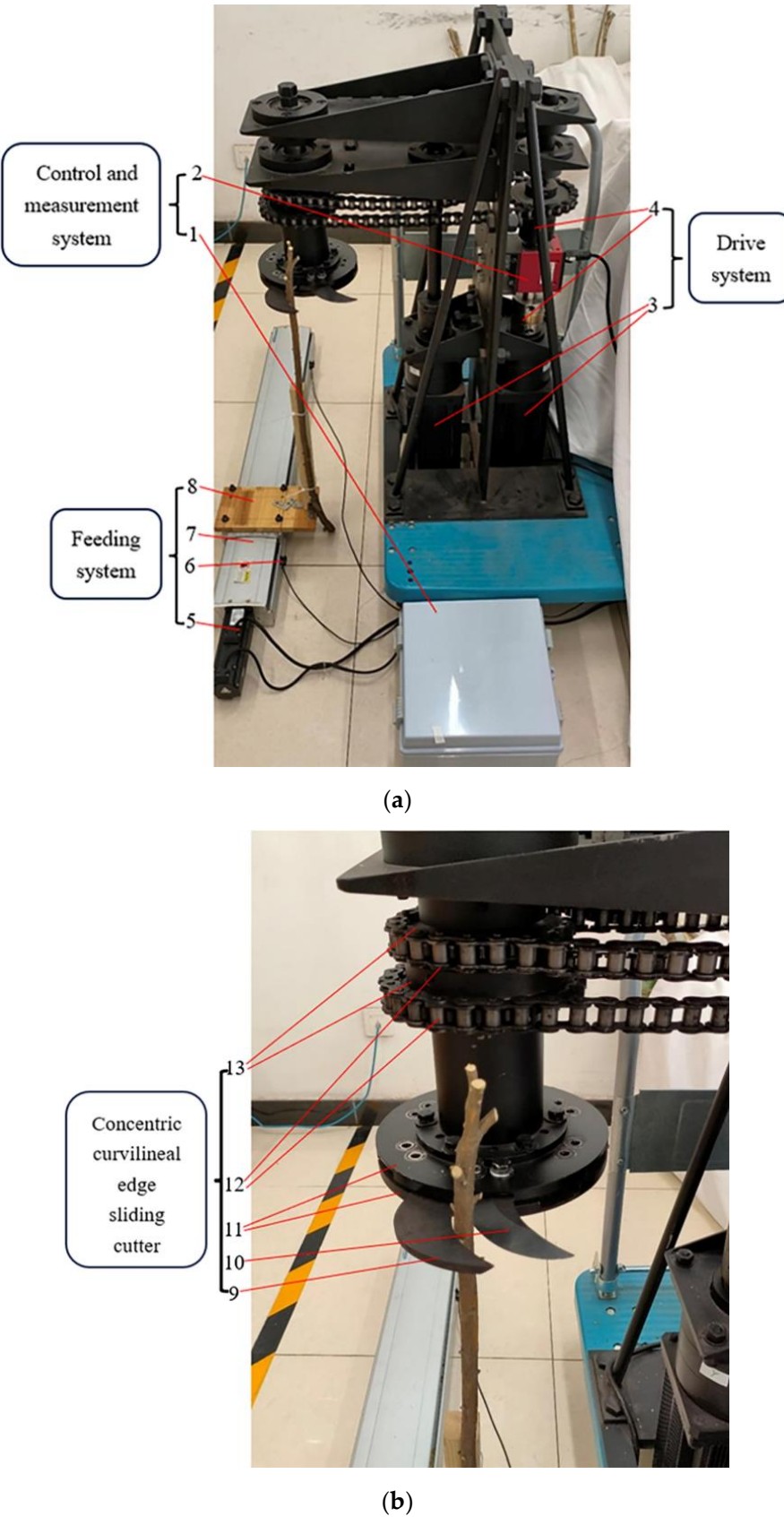

(**a**)

(**b**)

**Figure 2.** *Cont.*

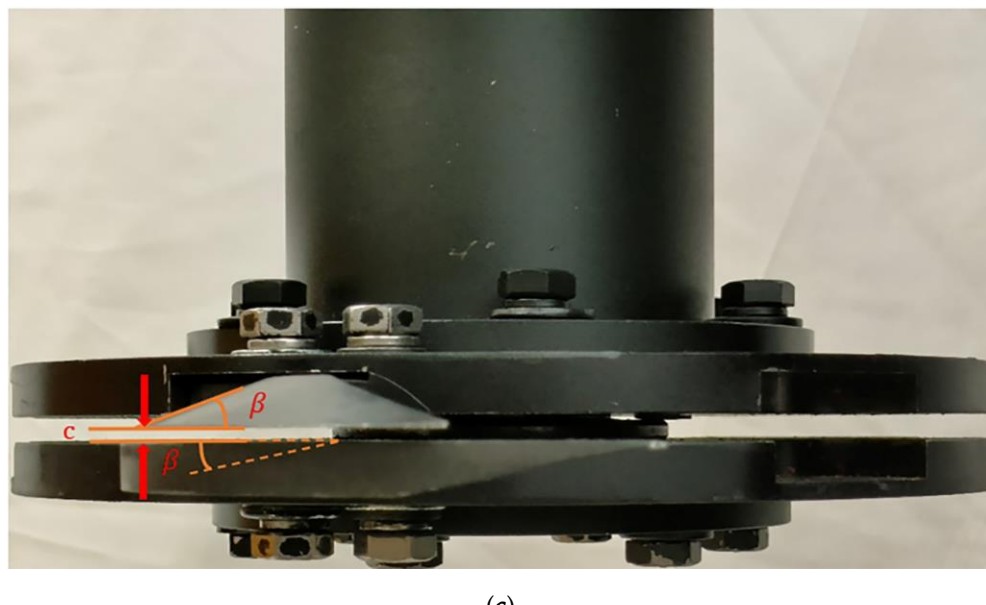

(**c**)

**Figure 2.** The cutting test bench. (**a**) The integral structure of cutting test bench; (**b**) Concentric curvilineal edge sliding cutter; (**c**) Cutting clearance and the wedge angle of the blade. In (**b**): 1. Control box (includes motion control card, motor drivers and power supply); 2. DYN-200 Dynamic torque sensor; 3. 130BYG350D Stepper motors and stepper motor reducers; 4. Couplings; 5. 60ASM400 AC Servo motor and reducer; 6. Limit sensors; 7. Moving guide rail; 8. Branch holder plate; 9. Curvilineal edge blade (fixed blade); 10. Curvilineal edge blade (moving blade); 11. Cutterheads; 12. Chains; 13. Chain wheels.

*2.4. Cutting Principle*

2.4.1. Clamping Stage

The cutting part consists of two curvilineal edge blades. The lower blade is used as a fixed blade, and the upper blade serves as a moving blade for cutting work. Before the cutting occurs, the two blades have a clamping stage for the branch, as shown in Figure 3.

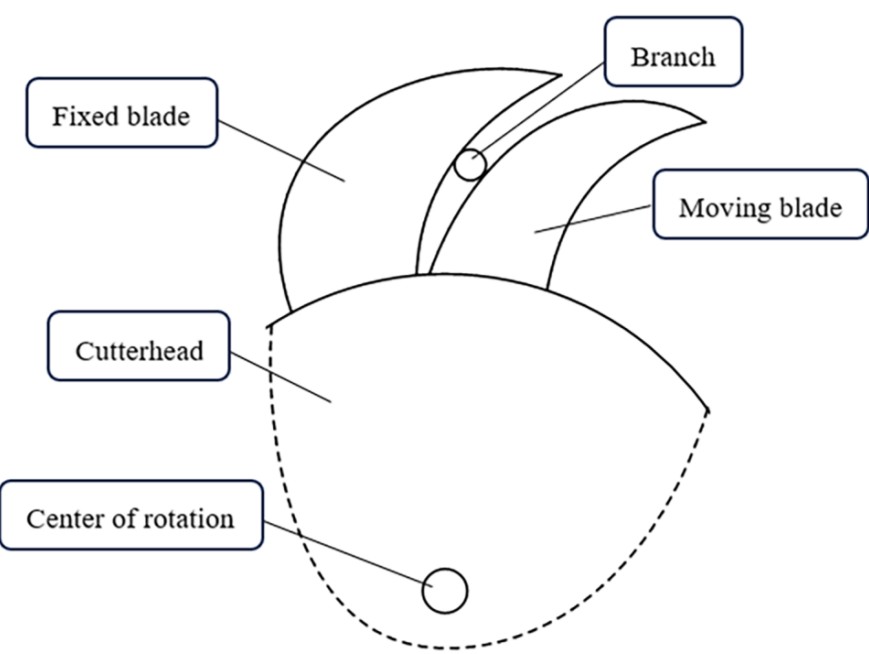

**Figure 3.** Clamping stage of concentric curvilineal edge sliding cutter.

In the clamping stage, the forces between the fixed blade, the moving blade and the branch are as shown in Figure 4; the branch makes contact with the fixed blade at point A, and makes contact with the moving blade at point B. The branch is restricted to a friction force $F_{fb}$ in the tangential direction and a support force $F_{Nb}$ in the normal direction at point B. The branch gives the fixed blade a friction force $F_{fa}$ in the tangential direction and a support force $F_{Na}$ in the normal direction at point A. The angle between the resultant force $F_b$ and support force $F_{Nb}$ is called the transmission angle; the transmission angle at point B is $\gamma_b$. The angle between the resultant force $F_a$ and support force $F_{Na}$ is called the friction angle; the friction angle at point A is $\varphi_a$.

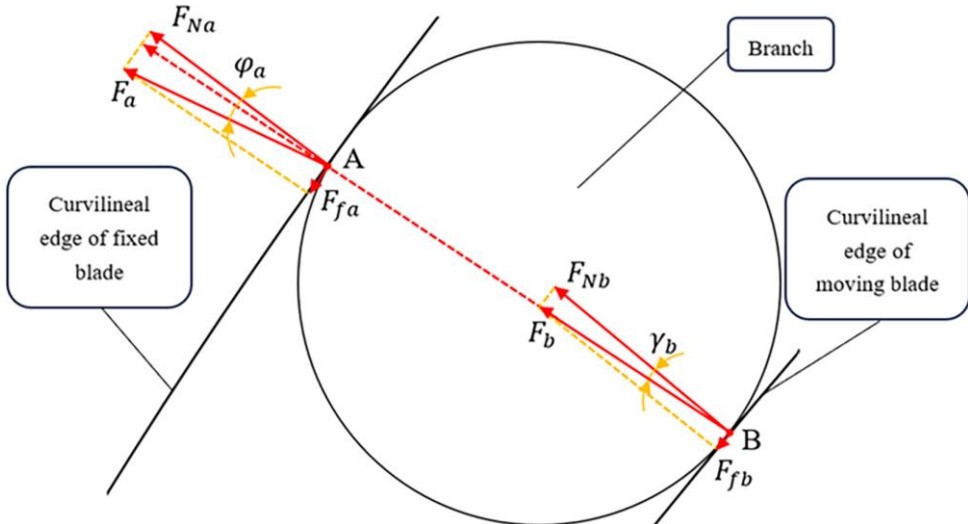

**Figure 4.** Mechanical model of Curvilineal edge of fixed blade and moving blade and branch in clamping stage.

As shown in Figure 4, the following conditions need to be met in order to ensure that the clamping is self-locking:

$$F_{fa} > F_{fb} \tag{2}$$

From the geometric relationship:

$$\varphi_a > \gamma_b \tag{3}$$

### 2.4.2. Sliding Stage

Sliding cutting is more labor-saving than straight-across cutting Sliding cutting refers to the cutting method in which the absolute motion direction of the cutter is neither vertical nor parallel to the cutting edge of the cutter [3]. The angle between the actual cutting direction of the cutter and the vertical direction of the edge is called the sliding cutting angle. The angle between the resultant force of friction and the support force is called the friction angle. In Figure 5, the sliding cutting angle is $\tau$ and the friction angle is $\varphi$. The basic condition of sliding cutting is that the friction angle is less than the sliding cutting angle [25]. Therefore, when the friction angle is less than the sliding cutting angle, it can be considered that the concentric curvilineal edge sliding cutter is of the sliding cutting type.

For the sliding stage, in Figure 5, for the actual cutting process, when the sliding cutting occurs at the middle point, the following blades are the main cutting blades for analysis. The normal force $F_N$ and the tangential force $F_T$ act on the branches at the same time. The blades move along their resultant force $F$ direction. The moving distance of the blade in the vertical direction is R, and the moving distance in the horizontal direction is S. Because there is friction $F_f$ between the blade and the shrub stem during cutting, the shrub branches slide in the horizontal direction relative to the blade under the action of

friction, and the sliding distance is $S_0$. Where $\varphi$ is the friction angle and $\tau$ is the sliding cutting angle, then the two angles can be expressed as:

$$an\tau = \frac{S}{R} \tag{4}$$

$$tan\varphi = \frac{S - S_0}{R} \tag{5}$$

The condition of sliding cutting is that the friction angle is less than the sliding cutting angle. Then there are:

$$S > S - S_0 \tag{6}$$

When the friction angle is always greater than zero, the sliding distance $S_0$ of shrubs under the action of friction is always greater than zero. It shows that sliding cutting occurs with the concentric curvilineal edge sliding cutter.

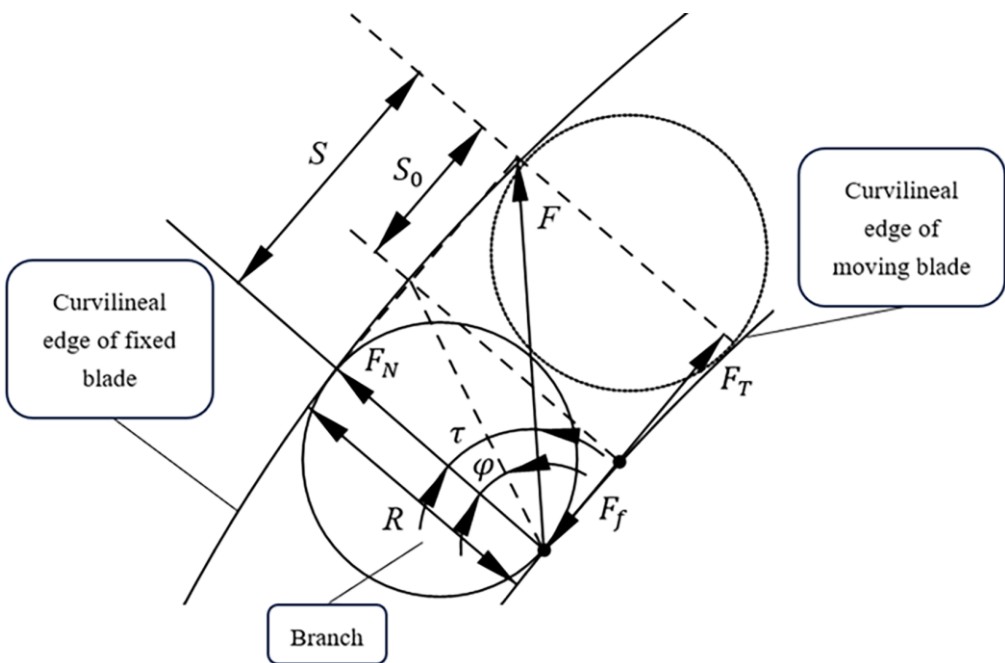

**Figure 5.** Sliding stage of concentric curvilineal edge sliding cutter.

## 3. Experimental Design

### 3.1. Single-Factor Experiment Design

A single-factor experimental design was planned. The branch diameter D, moisture content W, wedge angle β, cutting speed $V_c$ and cutting clearance c are taken as the factors for the target values of peak torque T and cutting energy E. The factors and levels of the single-factor test are shown in Table 1. In the C.K. stubble retention period, the diameter of most branches is generally less than 15 mm [2]. In the single-factor test, the diameter D was taken as 4–14 mm and 9–12 mm in other groups. The cutting speed was selected as 1.5–3.0 rad/s in the single-factor test to avoid operational risks, and 2.0 rad/s in the remaining groups. The wedge angle of the blade was selected from 20° to 50° in the single-factor test and 30° in other groups (Figure 6). When the cutting clearance is small, the friction between the blade and the branch is intensified, and the service life of the blade is affected. When cutting clearance is large, the cutting force increases sharply [16,26]. As a result, the range of the cutting clearance was 0.1 to 3 mm, and four levels were set. In the remaining groups, the cutting clearance was taken as 1 mm. After the branches were stored (22 ± 0.5 °C) for 5, 10, 15 and 20 days, the mean of moisture content of branches was 23.5%, 19.6%, 16.3% and 13.2%. The single-factor test of moisture content W was conducted across four different levels of moisture. In the remaining groups, fresh branches were used

as test objects (moisture content around 23%). In the cutting test bench, the upper blade is rotating and the lower blade is stationary. In order to ensure that the cutting position of the branch remains unchanged, the branch is close to the lower blade. The test conditions and variables are depicted in Table 1. Every group was tested five times, and the results are averaged.

**Table 1.** Factors and levels of the single-factor test.

| Level | Branch Diameter D (mm) | Cutting Speed $V_c$ (rad/s) | Wedge Angle $\beta$ (°) | Cutting Clearance c (mm) | Moisture Content W (%) |
|---|---|---|---|---|---|
| 1 | 4 | 1.5 | 20 | 0.1 | 13.2 |
| 2 | 6 | 2.0 | 30 | 1 | 16.3 |
| 3 | 8 | 2.5 | 40 | 2 | 19.6 |
| 4 | 10 | 3.0 | 50 | 3 | 23.5 |
| 5 | 12 | | | | |
| 6 | 14 | | | | |

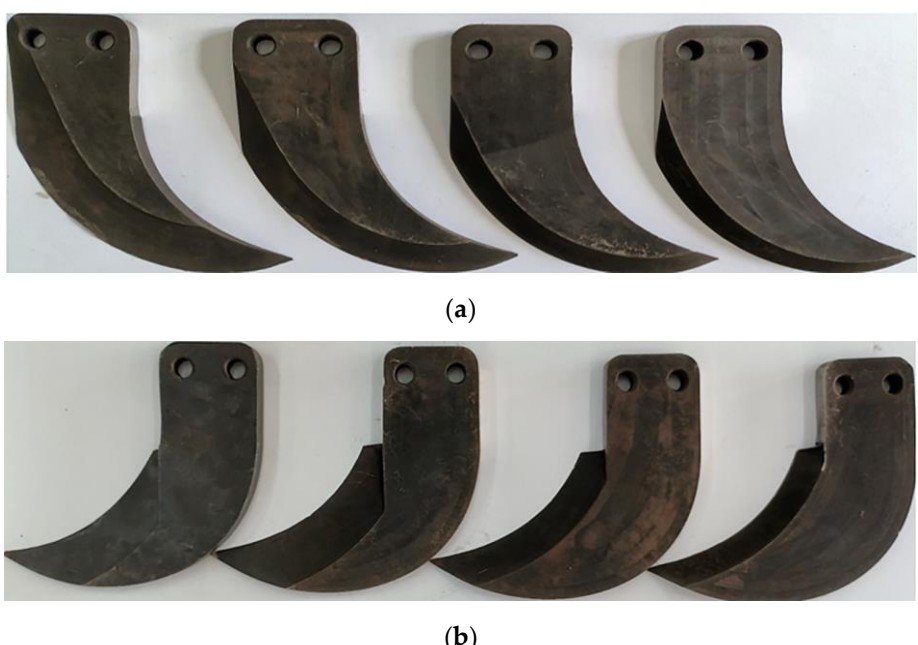

(a)

(b)

**Figure 6.** (**a**) The upper blade; the wedge angle of the blade from left to right is 20°, 30°, 40°, 50°. (**b**) The lower blade; the wedge angle of the blade from left to right is 20°, 30°, 40°, 50°.

### 3.2. Multi-Factor Experimental Design

Based on the Box—Behnken principle, a three-factor and three-level multi-factor test (Table 2) was planned, with cutting speed, wedge angle and cutting clearance as factors. Each group of tests was repeated five times to take the average value. The diameter of branches is 9~12 mm, and the moisture content is around 23%.

**Table 2.** Factors and levels of the multi-factor test.

| Level | Cutting Speed $V_c$ (rad/s) | Wedge Angle $\beta$ (°) | Cutting Clearance c (mm) |
|---|---|---|---|
| −1 | 1.5 | 20 | 1 |
| 0 | 2 | 30 | 2 |
| 1 | 2.5 | 40 | 3 |

## 4. Results

### 4.1. Single-Factor Test Results

The results of the single-factor test are depicted in Figures 7–11. The cutting energy rises with the growth of the branch diameter and the wedge angle. As the cutting clearance and cutting speed growth, the cutting energy decreases first and then increases. As the moisture content increases, the cutting energy gradually decreases.

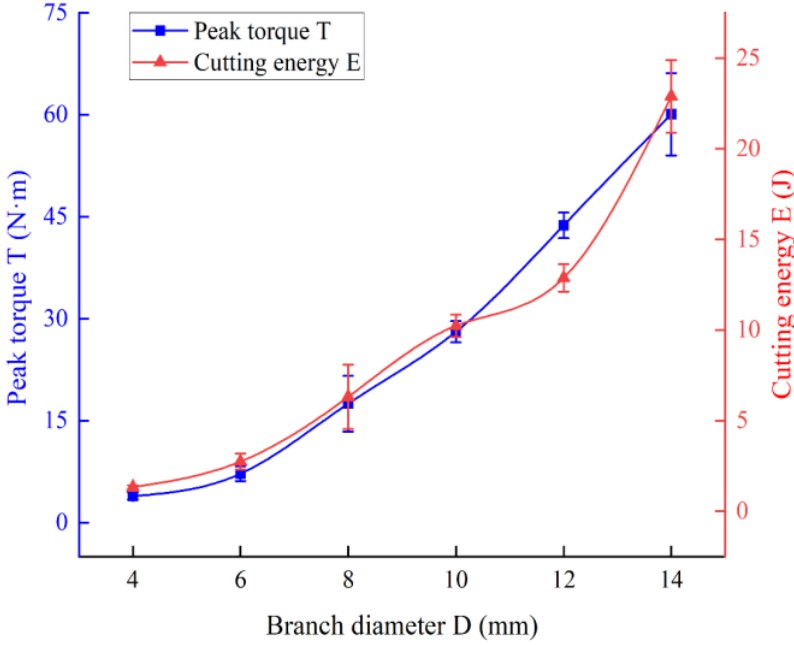

**Figure 7.** Cutting effect of different branch diameters. (Note: Cutting speed 2 rad/s; wedge angle 30°; cutting clearance 1 mm; moisture content around 23%).

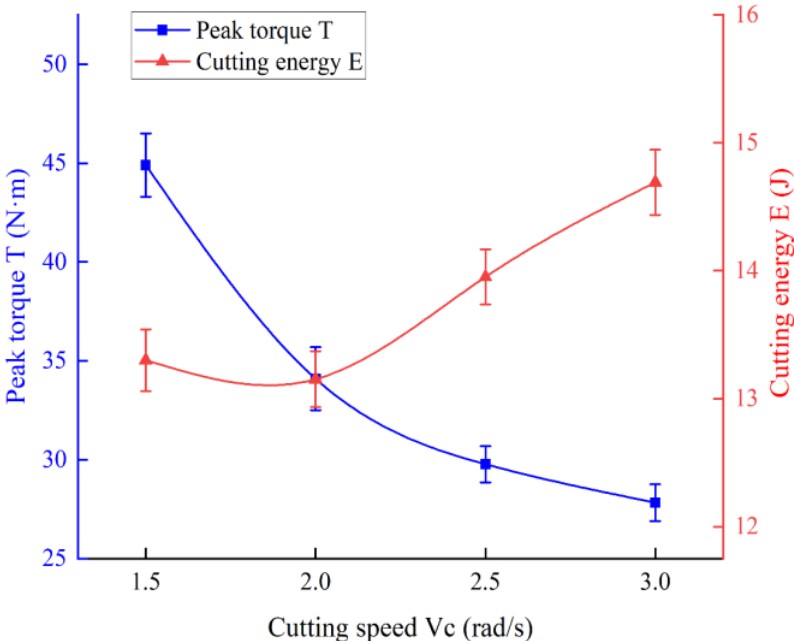

**Figure 8.** Cutting effect of different cutting speeds. (Note: Branch diameter 9~12 mm; wedge angle 30°; cutting clearance 1mm; moisture content around 23%).

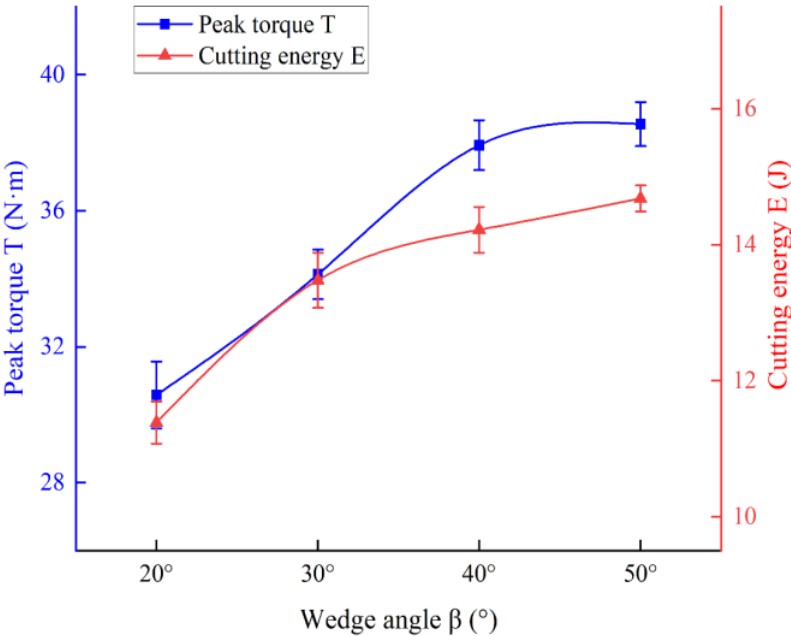

**Figure 9.** Cutting effect of different wedge angles. (Note: Branch diameter 9~12 mm; cutting speed 2 rad/s; cutting clearance 1 mm; moisture content around 23%).

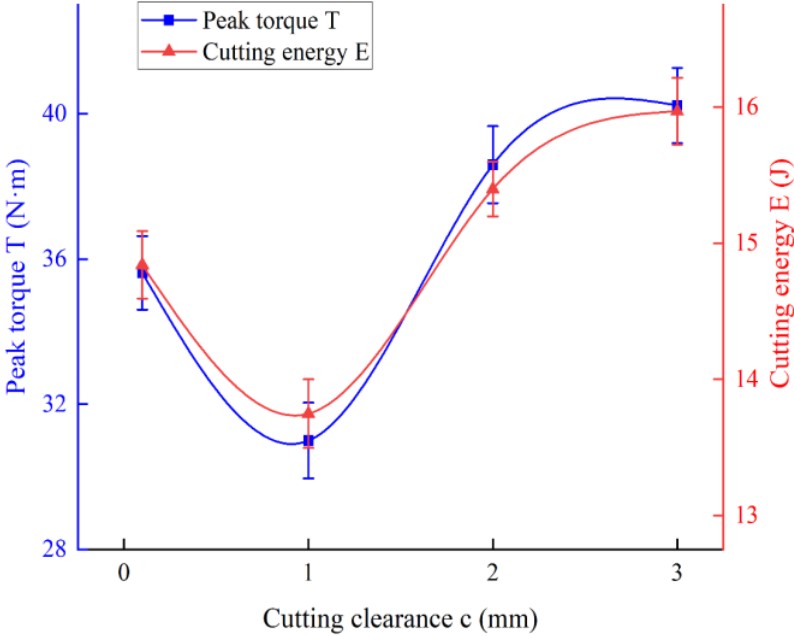

**Figure 10.** Cutting effect of different cutting clearances. (Note: Branch diameter 9~12 mm; cutting speed 2 rad/s; wedge angle 30°; moisture content around 23%).

The peak torque increases with the growth of branch diameter and wedge angle. With the growth of cutting speed and moisture content, the peak torque shows a trend of decreasing. The peak torque decreases first and then rises with the increase in cutting clearance.

### 4.2. Multi-Factor Test Results

After analyzing the results of the single-factor test, on the basis of the Box—Behnken principle, the multi-factor tests proceeded in random order, with a total of 12 experimental points and five central points to minimize the interference of external factors on the experiment. The test was repeated five times for each level and the average value is taken. The results of the tests are shown in Table 3 below.

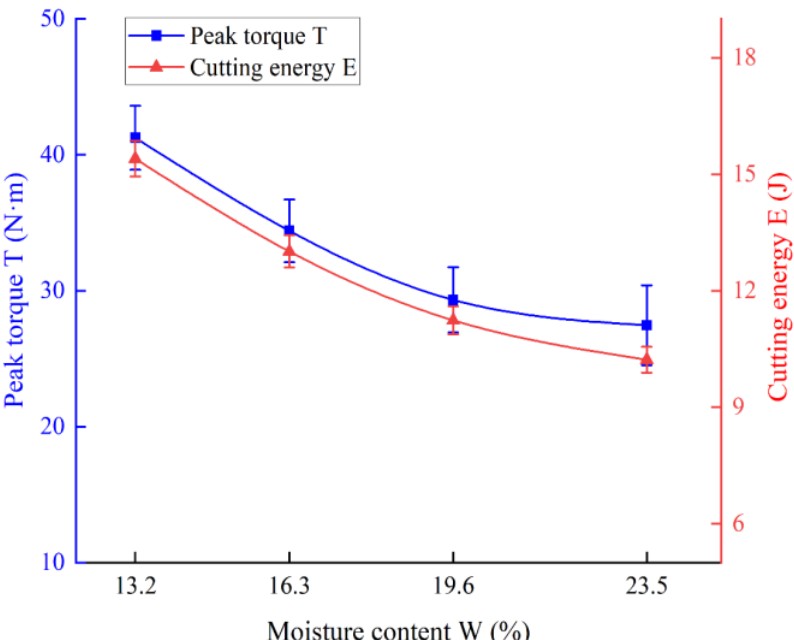

**Figure 11.** Cutting effect of different moisture contents. (Note: Branch diameter 9~12 mm; cutting speed 2 rad/s; wedge angle 30°; cutting clearance 1 mm).

**Table 3.** Design scheme for the cutting test and the results of the responses.

| Level | Cutting Speed $V_c$ (rad/s) | Wedge Angle $\beta(°)$ | Cutting Clearance c (mm) | Peak Torque T (N·m) | Cutting Energy E (J) |
|---|---|---|---|---|---|
| 1 | 1 | 0 | 1 | 18.90 | 8.50 |
| 2 | 0 | −1 | −1 | 17.83 | 6.98 |
| 3 | 0 | −1 | 1 | 19.12 | 7.54 |
| 4 | 1 | 0 | −1 | 17.56 | 8.00 |
| 5 | −1 | 0 | 1 | 23.42 | 8.18 |
| 6 | −1 | −1 | 0 | 22.30 | 6.54 |
| 7 | −1 | 0 | −1 | 22.20 | 7.18 |
| 8 | 1 | 1 | 0 | 20.24 | 8.68 |
| 9 | 0 | 0 | 0 | 19.97 | 8.06 |
| 10 | −1 | 1 | 0 | 24.23 | 7.78 |
| 11 | 0 | 0 | 0 | 19.74 | 8.07 |
| 12 | 0 | 0 | 0 | 20.26 | 8.09 |
| 13 | 0 | 0 | 0 | 19.86 | 8.00 |
| 14 | 1 | −1 | 0 | 17.25 | 7.24 |
| 15 | 0 | 0 | 0 | 20.14 | 7.87 |
| 16 | 0 | 1 | 1 | 22.30 | 8.80 |
| 17 | 0 | 1 | −1 | 21.06 | 8.16 |

By conducting these multi-factor tests, the results of the 12-group test and the five-group central test were obtained. The results of the five sets of center tests did not differ significantly from each other as expected due to the same test level settings. The other groups of tests differed considerably in the values of the test indicators due to the different settings of the test levels.

## 5. Discussion

### 5.1. Analysis of Single-Factor Tests

#### 5.1.1. Branch Diameter

As can be seen in Figure 7, the larger the diameter of the branch, the more cellulose will be cut, and the peak torque and cutting energy rise significantly [16,27]. All other things being equal, the larger the sectional area, the greater the friction between the blade and the branch. When the branch diameter exceeds the cutting limit of the tool, it will not be able to cut, and the blade will be jammed.

### 5.1.2. Cutting Speed

As shown in Figure 8, the peak torque decreases as the cutting speed rises. As cutting speed increases, the cutting energy decreases and then rises. C.K. branches are plant fibers that are composite, anisotropic and non-linear in nature. The cutting process is composed of two stages: extrusion deformation and cutting [28]. When the cutting speed is low, the branches undergo large compression deformation. As the cutting speed increases, the point of branch cutting moves faster and the cutting time decreases gradually. Additionally, the compression deformation of the branch caused by the blade decreases over time [16]. The branch transitions from the stage of extrusion deformation to the cutting stage, with a subsequent gradual reduction in peak torque and cutting energy. When the speed exceeds a certain value and continues to grow, the extrusion deformation time of the branch does not change significantly [29]. Therefore, the change trend of the peak torque is gradually smooth, and the cutting energy diminishes first and then rises with the increase in cutting speed.

### 5.1.3. Wedge Angle

In the cutting test, when the wedge angle is 20° to 40°, the branches can be cut smoothly at once. As illustrated in Figure 9, the peak torque and cutting energy increase as the wedge angle increases. It means that a smaller wedge angle can cut branches more easily. The value of the wedge angle determines the cutting edge's strength and affects the longevity of the blade [14]. Therefore, the wedge angle's impact on the blade's longevity should be taken into account.

### 5.1.4. Cutting Clearance

As the cutting clearance c increases, both peak torque and cutting energy exhibit a pattern of initially decrease and then subsequently increase. When c is 1 mm, the peak torque and cutting energy are the smallest. When the cutting clearance is too large or too small, the greater the absolute value of the difference between the shear band and the fracture band and the greater the peak torque and cutting energy. When c is 1mm, the shear band ratio and the fracture zone ratio are complementary, and the peak torque and cutting energy are the smallest. The results of the cutting process for various cutting clearances are displayed in Figure 10. Figure 12 illustrates the cutting sections at different cutting clearances. As the cutting clearance is reduced, the quality of the cutting section improves.

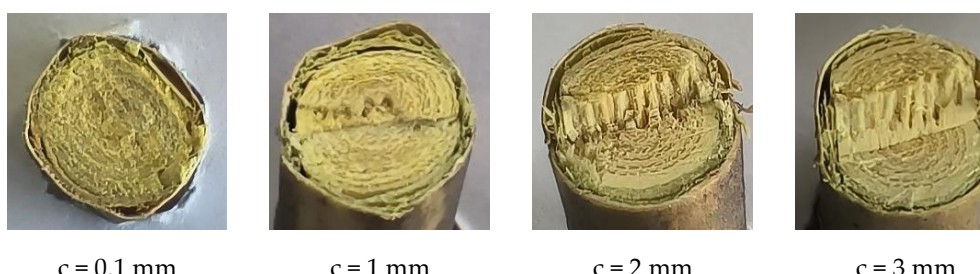

c = 0.1 mm　　　　　c = 1 mm　　　　　c = 2 mm　　　　　c = 3 mm

**Figure 12.** The cutting sections with different cutting clearance.

### 5.1.5. Moisture Content

The average moisture content of freshly-cut branches was 29.3%. Subsequently, the moisture content of branches stored for 5, 10, 15, and 20 days was measured and determined to be 23.5%, 19.6%, 16.3%, and 13.2%, respectively. Each of the four groups of branches, classified by their moisture content, underwent cutting tests.

During the cutting operation, the peak torque and cutting energy decrease with the increase in moisture content, as shown in Figure 11. As the moisture content of a branch decreases and the dry matter increases, its ultimate stress increases [16]. C.K. Branches are recommended to be harvested during the season when their moisture content is high.

*5.2. Multi-Factor Test Analysis*

5.2.1. Variational Analysis (ANOVA)

Based on the ANOVA results presented in Table 4, it is evident that $V_c$, $\beta$, c, $V_c^2$ have a statistically significant effect on peak torque ($p < 0.01$), whereas the other factors do not have a significant effect ($p > 0.05$). Additionally, the failure to fit term, $p = 0.2395$, is also not significant, indicating that there are no other major factors contributing to the test indicators.

**Table 4.** Variational analysis for peak torque.

| Variance Source | Sum of Squares | Degree of Freedom | Mean Square | F Values | *p* Values |
|---|---|---|---|---|---|
| Model | 63.73 | 9 | 7.08 | 109.04 | <0.0001 |
| $V_c$ | 41.4 | 1 | 41.40 | 637.64 | <0.0001 ** |
| $\beta$ | 16.05 | 1 | 16.05 | 247.11 | <0.0001 ** |
| c | 3.24 | 1 | 3.24 | 49.87 | 0.0002 ** |
| $V_c \cdot \beta$ | 0.2809 | 1 | 0.2809 | 4.33 | 0.0761 |
| $V_c \cdot c$ | 0.0036 | 1 | 0.0036 | 0.0554 | 0.8206 |
| $\beta \cdot c$ | 0.0006 | 1 | 0.0006 | 0.0096 | 0.9246 |
| $V_c^2$ | 2.22 | 1 | 2.22 | 34.25 | 0.0006 ** |
| $\beta^2$ | 0.3402 | 1 | 0.3402 | 5.24 | 0.0559 |
| $c^2$ | 0.1697 | 1 | 0.1697 | 2.61 | 0.1500 |
| Residual | 0.4545 | 7 | 0.00649 | | |
| Lack of Fit | 0.2794 | 3 | 0.00931 | 2.13 | 0.2395 |
| Pure Error | 0.1751 | 4 | 0.0438 | | |
| Cor Total | 64.18 | 16 | | | |

Note: $p < 0.01$ (highly significant, **).

Design-expert 11.0 software was used to analyze the test results. Multiple regression fitting was performed for each test index to eliminate insignificant factors and obtain the regression formula for each factor level on the peak torque, as follows:

$$T = -2.27V_c + 1.42\beta + 0.6363c + 0.7314V_c^2 + 20.03 \tag{7}$$

Table 5 shows the ANOVA for the cutting energy E. For the cutting energy, the main order of influence is $V_c$, $\beta$, c, $V_c^2$, $\beta^2$, $c^2$; with the effects of $V_c$, $\beta$, c, $V_c^2$, $\beta^2$ being of great significance ($p < 0.01$), $c^2$, $V_c \cdot c$ being significant ($0.01 < p \leq 0.05$), while the effects of the other factors were not found to be significant ($p > 0.05$). The failure to fit term, $p = 0.2648$, is insignificant, indicating that no additional test indicators were influenced by external factors.

Design-expert 11.0 software was used to analyze the test results, multiple regression was fitted to each of the test metrics and insignificant factors were excluded to obtain the following regression equations for the level of influence of each factor on the cutting energy:

$$E = 0.3425V_c + 0.64\beta + 0.3375c - 0.125V_c \cdot c0.1815V_c^2 - 0.2765\beta^2 + 0.1285c^2 + 8.02 \tag{8}$$

**Table 5.** Variational analysis for cutting energy.

| Variance Source | Sum of Squares | Degree of Freedom | Mean Square | F Values | *p* Values |
|---|---|---|---|---|---|
| Model | 5.73 | 9 | 0.6366 | 58.04 | <0.0001 |
| $V_c$ | 0.9385 | 1 | 0.9385 | 85.56 | <0.0001 ** |
| $\beta$ | 3.28 | 1 | 3.28 | 298.74 | <0.0001 ** |
| c | 0.9112 | 1 | 0.9112 | 83.08 | <0.0001 ** |
| $V_c \cdot \beta$ | 0.0100 | 1 | 0.0100 | 0.9117 | 0.3715 |
| $V_c \cdot c$ | 0.0625 | 1 | 0.0625 | 5.70 | 0.0484 |
| $\beta \cdot c$ | 0.0016 | 1 | 0.0016 | 0.1459 | 0.7138 |
| $V_c^2$ | 0.1387 | 1 | 0.1387 | 12.65 | 0.0093 ** |
| $\beta^2$ | 0.3219 | 1 | 0.3164 | 29.35 | 0.0010 ** |
| $c^2$ | 0.0695 | 1 | 0.0695 | 6.34 | 0.0399 * |
| Residual | 0.0768 | 7 | 0.0110 | | |
| Lack of Fit | 0.0449 | 3 | 0.0150 | 1.88 | 0.2743 |
| Pure Error | 0.0319 | 4 | 0.0080 | | |
| Cor Total | 5.81 | 16 | | | |

Note: $p < 0.01$ (highly significant, **); $p < 0.05$ (significant, *).

### 5.2.2. Response Surface Analysis

According to Figure 13, the influence of cutting speed, wedge angle and cutting clearance on the peak torque are highly significant. The results of the regression equation shown in Table 4 are consistent with the trend observed in the response surface. The total impact law between the factors is as follows: the peak torque increases with the growth of wedge angle and cutting clearance, and the peak torque rises with the reduction in cutting speed, which is identical to the result obtained using the single-factor test.

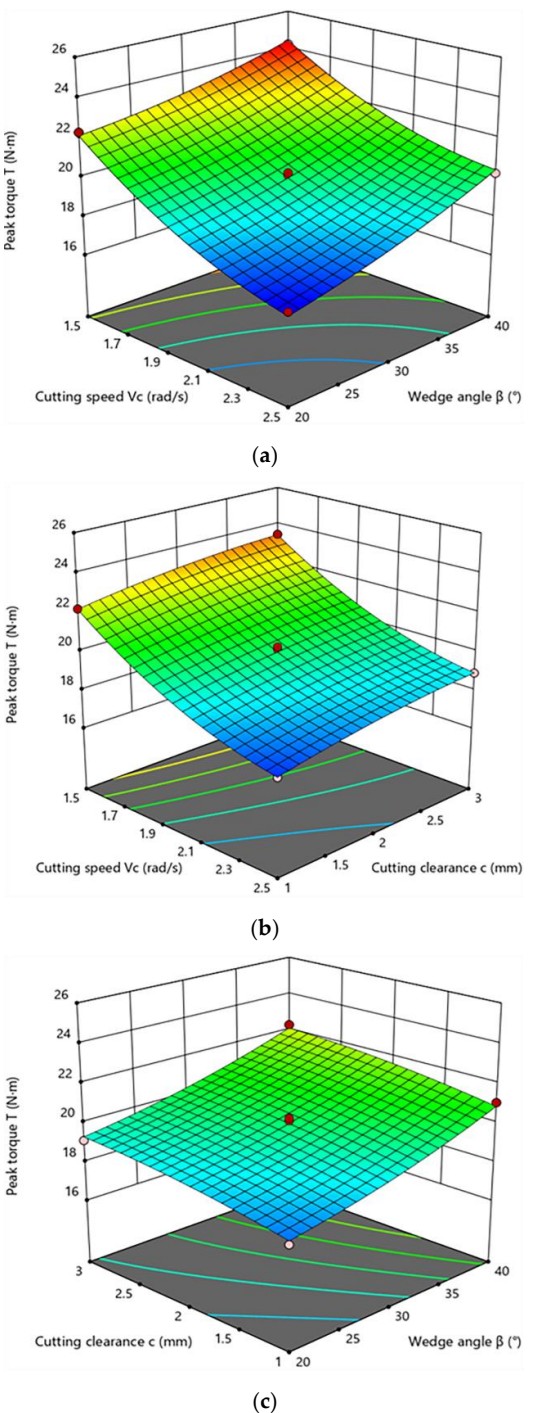

**Figure 13.** The influence of interaction factors on peak torque. (**a**) $T = f(V_c, \beta, 0)$; (**b**) $T = f(V_c, 0, c)$; (**c**) $T = f(0, \beta, c)$.

As illustrated in Figure 14, the trend observed in the response surface for cutting energy is identical to the ANOVA regression results presented in Table 5. The total influence of each factor is that the consumption of cutting power increases as the cutting speed, the angle of the wedge and the clearance of the cut increase.

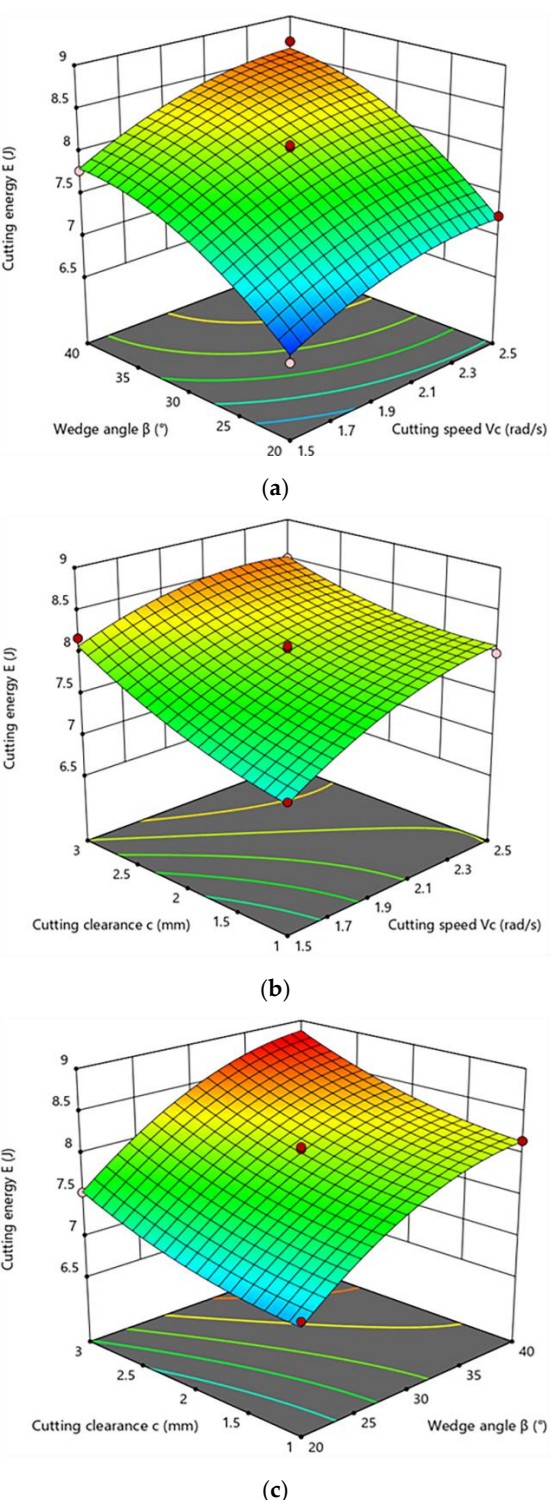

**Figure 14.** The influence of interaction factors on cutting energy. (**a**) $P = f(V_c, \beta, 0)$; (**b**) $P = f(V_c, 0, c)$; (**c**) $P = f(0, \beta, c)$.

*5.3. Optimization of the Parameters and Experimental Validation*

In order to obtain the best combination of performance parameters, the minimum values of peak torque and cutting energy have been used as the targets for optimization. The analysis is based on the operating conditions and outcomes; optimal solution conditions include the following:

$$\begin{cases} 1.5 \text{ rad/s} \leq V_c \leq 2.5 \text{ rad/s} \\ \quad 20° \leq \beta \leq 40° \\ \quad 1 \text{ mm} \leq c \leq 3 \text{ mm} \\ minT = f_1(V_c, \beta, c) \\ minP = f_2(V_c, \beta, c) \end{cases} \tag{9}$$

To optimize the solution, a set of optimal cutting parameter combinations and their target values were obtained by using Design-expert 11.0 software. The cutting speed is 2.16 rad/s, the wedge angle is 20°, the cutting clearance is 1.0 mm, the predicted values of peak torque and cutting energy are 17.25 N·m and 7.03 J. Based on the optimal parameters, the average peak torque and cutting energy measured on the cutting test bench were 17.4 N·m and 6.68 J. The discrepancies between the observed and forecasted values for T and P are 0.87 and 5.004 percent (Table 6).

**Table 6.** Analysis of variance for peak torque and cutting energy.

| Test Serial Number | Peak Torque T (N·m) | Cutting Energy E (J) |
|:---:|:---:|:---:|
| 1 | 17.5 | 6.72 |
| 2 | 17.2 | 6.54 |
| 3 | 16.8 | 6.31 |
| 4 | 18.2 | 7.21 |
| 5 | 17.3 | 6.62 |
| Average value | 17.4 | 6.68 |
| Relative errors | 0.87% | 5.004% |

*5.4. Comparison of Peak Cutting Force and Cutting Energy*

The optimal parameter combination and its peak torque and cutting energy are obtained. The model's accuracy is confirmed through empirical testing, and a set of actual experimental data is obtained. The peak torque and cutting energy under the optimal parameter combination are 17.4 N·m and 6.68 J, the actual cutting radius is 120~130 mm and the corresponding peak cutting force can be calculated from the Equation (10).

$$F = \frac{M}{R} \tag{10}$$

The peak cutting force is calculated to be 135.75~141.35 N; the mean value is 138.55 N. The actual *F* value is 138.55 N and *E* value is 6.68 J. The cutting experiment of C.K. branches was carried out by using a reciprocating cutter. In contrast to paper [5], a set of optimal parameter combinations for cutting C.K. branches was obtained through the reciprocating cutter; under the optimal parameter combination, the peak cutting force is 654.14 N and the cutting energy is 6.20 J. Thus, the peak cutting force of this concentric curvilineal edge sliding cutter is reduced by 78.8%. In another typical paper [30], through a self-made reciprocating cutting test bench, a set of optimal parameter combinations for cutting apple branches was obtained; the peak cutting force was 560.97 N with the optimal combination of parameters. In comparison, the peak cutting force of the concentric curvilineal edge sliding cutter is reduced by 75.3%. The concentric curvilineal edge sliding cutter is proved to be more labor-saving with its sliding cutting characteristics.

## 6. Conclusions

1.  A concentric curvilineal edge sliding cutter is proposed in this paper. Two sets of curvilineal edge blades were mounted on two concentrically nested cutterheads. The clamping stage and sliding stage of the concentric curvilineal edge sliding cutter were studied to explore the cutting mechanism;

2.  With the concentric curvilineal edge sliding cutter, the peak torque and cutting energy of C.K. branch pruning stubble were tested and studied. Through single-factor and multi-factor experiments, the relationship between branch diameter, cutting speed, wedge angle, cutting clearance, moisture content and peak torque, cutting energy were explored. Test results: (1) Peak torque grows with the increase in branch diameter and wedge angle; it decreases with the growth of cutting speed and moisture content, and with the growth of cutting clearance, it declines first and then rises. (2) Cutting energy increases with increasing branch diameter and wedge angle, it reduces with the increase in moisture content, and as the growth of cutting clearance and cutting speed, first it will reduce and then it will rise. (3) The results of multi-factor tests were consistent with those of single-factor tests. The optimum combination of parameters is the following: cutting speed is 2.16 rad/s, wedge angle is 20°, cutting clearance is 1.0 mm. The peak torque for this combination is 17.25 N·m, and the cutting energy is 7.03 J, which are confirmed by the validation tests with discrepancies of 0.87% and 5.004%.

3.  Compared with the reciprocating cutting tool under the optimal parameter combination, the peak cutting force is reduced by 78.8%. The concentric curvilineal edge sliding cutter is verified to be more labor-saving with guaranteed cutting section quality and a very similar cutting energy. The concentric curvilineal edge sliding cutter can be used as a cutting tool module to provide rotational and forward momentum to support it in its work. It can provide new cutter and data support for the development of subsequent C.K. branch stubble equipment.

**Author Contributions:** Conceptualization, S.G., H.L. and J.K.; methodology, S.G., H.L. and J.K.; software, S.G. and Z.Z.; validation, S.G. and Z.Z.; formal analysis, S.G.; investigation, S.G.; resources, H.L. and J.K.; data curation, S.G. and Z.Z.; writing review and editing, S.G. and H.L.; visualization, S.G. and Z.Z.; supervision, H.L. and J.K.; project administration, S.G. and Z.Z. All authors have read and agreed to the published version of the manuscript.

**Funding:** National Natural Science Foundation of China (Grant number: 51705022) and Enterprise Projects 2023HXKFGXY002.

**Institutional Review Board Statement:** Not applicable.

**Informed Consent Statement:** Not applicable.

**Data Availability Statement:** The data presented in this study are available on request from the corresponding author.

**Acknowledgments:** Sincere thank also goes to Yaya Gao for his kind help in experimental materials.

**Conflicts of Interest:** The authors declare no conflict of interest.

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
