# Peer review of "Sliding Cutting and Cutting Parameters of Concentric Curvilineal Edge Sliding Cutter for Caragana korshinskii (C.K.) Branches"

_forests, doi:10.3390/f14122379_

Round 1

Reviewer 1 Report

Comments and Suggestions for Authors

The presented article deals with the interesting issue of pruning the branches of Caragana korshinckii (C.K.) plants, whose stands play an important role in strengthening soil surfaces in the sandy areas of northwestern China. This unpretentious plant thus fulfills an important anti-erosion role, but it is also a source of biomass that can be used by local residents. In order for the plant (C.K.) to fulfill the expected effects, its branches must be trimmed regularly. The authors of the article present the results of their study, which brings new possibilities for the implementation of branch pruning (C.K.), which are more productive compared to existing pruning methods, provide a better quality cut and are also less energy demanding. The article brings new interesting knowledge about the issue and the information contained in it can be used not only in connection with pruning (C.K.), but also other types of wood.

Chapter 1 – Introduction provides in-depth information about (C.K.), its bionomics, its significance for the given regions of China, as well as the characteristics of the current methods of pruning its branches (e.g. one-way, reversible, with or without support), while evaluating the advantages and disadvantages of these methods of cutting. Based on the analysis of existing cutting methods, the authors of the article created a new principle of the cutting device, which they verified experimentally during their study and assessed the effects of individual parameters of the cutting device on the overall result with statistical tests. I have quite fundamental comments about chapter 1: compared to usual practices, this chapter is disproportionately extensive, and therefore I recommend that it be significantly shortened. Some of its parts can possibly be partially used in other chapters of the article.

Chapter 2 – Materials and Methods describes the properties, origin and method of basic treatment of the tested branch material (C.K.). Furthermore, the test cutting bench is described in detail, while the text sections are appropriately supplemented with relevant images. The description of the principle of the test bench is thus easy to understand.

Chapter 3 – Test Indicators consists of two parts. In the first of them, the authors deal with the instrumental solution of detecting the peak torque of the new principle of the cutting mechanism developed by them. In the second part of this chapter, the principle of determining cutting power consumption values is analyzed and the basic mathematical equation that characterizes it is created. In the third, most extensive part of chapter 3, a detailed description of the functional principle of the construction of the cutting mechanism is given, an important part of which are the cutting jaws: one fixed and the other movable. This is considered by the authors to be the main innovation factor of their research solution. The description of the cutting device is comprehensible, even in this case illustrative diagrams of the cutting device are provided, facilitating the reader's understanding of the function of the new principle of the cutting device. From a scientific point of view, it is very positive that the authors have the design solution of their experimental cutting device supported theoretically by means of models and mathematical equations.

Chapter 4 – Experimental Design describes two principles of experiments in two subsections: single-factor and multi-factor design. As part of the single-factor experiments, the effects of a set of different knives of the cutting device, which differed from each other in the angles of their blades, were investigated. In the multifactorial research, other variables such as cutting speed, wedge angle or cutting clearnce were taken into account. It is obvious that the authors for their experimental investigation of the properties of the cutting mechanism have chosen suitable quantities, which they can use to demonstrate the optimal state of the given design solution of the cutting mechanism.

In chapter 5 – Results, significant findings from the research solution of the new principle of cutting device for plant branches (C.K.) are presented. These findings are first presented in subsection 5.1, which deals with the results of single-factor tests. The knowledge here is characterized by brief texts, supplemented by illustrative graphs. It can be concluded that the findings presented are apparently factually correct, also because some of them are partially predictable (two examples for all: even with other principles of cutting/cutting branches, it is known that the peak torque is directly dependent on the branch diameter or that increased wood moisture reduces the cutting resistance overcome by cutting tools). Subchapter 5.2 describes the results of multi-factor tests. In this case, all knowledge about these tests is entered in a single table (Tab. 3), but without further comment. I would recommend that a brief comment be added here.

Chapter 6 – Discussion assesses the findings from the individual tests in a suitable form. I think that this chapter is prepared correctly, it objectively assesses the established facts, and I am happy to state that the authors argue about their knowledge even in relation to the knowledge of other authors. This completes the purpose of the Discussion chapter. In chapter 5, the authors, among other things, use the analysis of multi-factor tests using the ANOVA application. Perhaps it would be more appropriate to mention this passage already in the previous subsection 5.2 (see my note above). Furthermore, I would like to emphasize that within this chapter 6, the authors also developed a formulation of the optimal parameters of the cutting device, which I consider to be a particularly positive fact, because the prerequisites are thus created for the eventual application of the newly proposed design principle of the cutting device for trimming plant branches (C.K.). Furthermore, I believe that it would be useful if the authors would at least indicate the method of applying the new cutting device on a real, operationally usable machine.

Chapter 7 – Conclusions summarizes the knowledge presented in the previous chapters of the article in a standard way. I have no comments about her.

References are written in a standard way, they contain the works of renowned authors, and their scope testifies to the conscientious preparation of the authors during the implementation of the research study and the article itself.

In conclusion, I state that the submitted article presents new original information, is processed based on the scientific approach of the authors, and its composition meets the requirements for scientific articles.

Reviewer 2 Report

Comments and Suggestions for Authors

Authors to realize the reduction of cutting force and guarantee pruning section quality in the pruning and stubble work of Caragana korshinskii (C.K.), a concentric curvilineal edge sliding cutter proposed and studied the cutting process by using this new sliding cutting tool.

Authors should make some changes and additions to the text.

1. In the abstract Authors wrote: "... The optimal parameter combination of the regression model was obtained with ?? of 2.16346 rad/s, ? of 20° and ? of 1.0mm, which resulted in the T of 17.25 N·M and P of 7.0319 J. The discrepancies between the observed and forecasted values for T and P are 0.87 and 5.004 percent. Compared with the reciprocating cutting tool, the peak cutting force of the tool is reduced by 78.8% under the optimal parameter combination..." There is too much details. Please correct it.

2. The structure of this paper is different like "in standard" papers. The section 3. Test Indicators should be removed to the section 2.

3. More properties of Caragana korshinskii is necessary.

4. The curves on the Figures 7-11 with the mathematical formulas with R2 coefficient can be described. It will be better for deeper analysis.

5. How many branches were selected for the experiments? How many repetitions were performed?

6. In the Table 3, the power consumption of the cut is given to the precision of 3-4 places. Is such a high precision necessary? Why? Please correct it.

7. The description of the axes in Figures 13-14 is poor. Please improve their quality.

Reviewer 3 Report

Comments and Suggestions for Authors

I have conducted a thorough evaluation of the manuscript titled "Sliding Cutting and Cutting Parameters of Concentric Curvilinear Edge Sliding Cutter for Caragana korshinskii Branches." Firstly, I would like to express my sincere appreciation for your significant contribution to advancing research in the optimization of sliding cutting technology. Your study is both fascinating and holds considerable potential for application in both academic and practical fields. The methodology, study design, and conducted experiments are notably praiseworthy. Additionally, the presentation of results is clear and well-structured. However, the manuscript requires improvements in its writing and analysis. To this end, I offer the following suggestions and corrections for your consideration:

1)      The term “Caragana korshinskii” is excessively emphasized throughout the manuscript, such as in the title, abstract, and introduction. This seems unnecessary, as the paper's primary focus is on cutting technology rather than the specifics of the shrub species. It would be advisable to commence the introduction with a review of relevant literature and a discussion on the concepts of cutting tools, omitting the first paragraph that currently opens the section.

2)      The introduction currently intertwines literature review and the author's work in a manner that lacks clarity. I recommend clearly separating these two components to enhance readability and distinction.

3)      Overall improvements in the manuscript's writing are necessary. For instance, sentences like “The cross-section quality is determined by the quality of the cross section., and Based on knowledge of different cutting types, cutting methods and the characteristics of various cutting tools.” are ambiguous. Additionally, it is advisable to remove the bracketed abbreviation (C.K.) after its initial definition for clarity.

4)      There appears to be an inconsistency with Equation 1, which seems to represent an energy equation rather than power. Moreover, there is a discrepancy in the definition of 'M' between the abstract, where it is defined as moisture content, and in Equation 10, where it corresponds to peak torque. This needs verification and correction.

5)      Please ensure the accuracy of the dimensions and units used, such as using N.m instead of N.M., and recognizing that Joules (J) are units of energy, while power is measured in watts (W = J/s). Similarly, images 7-11 and the tables require revision and updating.

Comments on the Quality of English Language

Overall improvements in the manuscript's writing are necessary. For instance, sentences like “The cross-section quality is determined by the quality of the cross section., and Based on knowledge of different cutting types, cutting methods and the characteristics of various cutting tools.” are ambiguous. Additionally, it is advisable to remove the bracketed abbreviation (C.K.) after its initial definition for clarity.
